# Implementation of national antenatal hypertension guidelines: a multicentre multiple methods study

Rebecca Whybrow ![ORCID],[1,2] Louise Webster,[1] Joanna Girling,[3] Heather Brown,[4] Hannah Wilson,[1] Jane Sandall,[1] Lucy Chappell[1,2]

¹Women and Children's Health, King's College London, London, UK
²Division of Women and Children's Health, Guy's and Saint Thomas' Hospitals NHS Trust, London, UK
³Women's Health Service, Chelsea and Westminster Hospital NHS Foundation Trust, London, UK
⁴Maternity, Brighton and Sussex University Hospitals NHS Trust, Brighton, UK

**Correspondence to**
Rebecca Whybrow;
rebecca.whybrow@kcl.ac.uk

## ABSTRACT

**Objective** To evaluate the implementation of National Institute for Health and Care Excellence antenatal hypertension guidelines, to identify strategies to reduce incidences of severe hypertension and associated maternal and perinatal morbidity and mortality in pregnant women with chronic hypertension.

**Methods** We used a multiple method multisite approach to establish implementation of guidelines and the associated barriers and facilitators. We used a national survey of healthcare professionals (n=97), case notes review (n=55) and structured observations (n=42) to assess implementation. The barriers and facilitators to implementation were identified from semistructured qualitative interviews with healthcare professionals (n=13) and pregnant women (n=18) using inductive thematic analysis. The findings were integrated and evaluated using the Consolidated Framework for Implementation Research.

**Setting and participants** Pregnant women with chronic hypertension and their principal carers (obstetricians, midwives and physicians), at three National Health Service hospital trusts with different models of care.

**Results** We found severe hypertension to be prevalent (46% of case notes reviewed) and target blood pressure practices to be suboptimal (56% of women had an antenatal blood pressure target documented). Women were infrequently given information (52%) or offered choice (19%) regarding antihypertensives. Women (14/18) reported internal conflict in taking antihypertensives and non-adherence was prevalent (8/18). Women who were concordant with treatment recommendations described having mutual trust with professionals mediated through appropriate information, side effect management and involvement in decision making. Professionals reported needing updates and tools for target blood pressure setting and shared decision making underpinned by antihypertensive safety and effectiveness research.

**Conclusions** Women's non-adherence to antihypertensives is higher than anticipated. Suboptimal information provision around treatment, choice of antihypertensives and target setting practices by healthcare professionals may be contributory. Understanding the reasons for non-adherence will inform education and decision-making strategies needed to address both clinician and women's behaviour. Further research into the effectiveness and long-term safety of common antihypertensives is also required.

## Strengths and limitations of this study

► Multiple methodological approaches and an implementation framework improved the reliability, validity and generalisability of the study.
► Structured observations were carried out using a validated tool with high inter-rater reliability.
► Women's medication behaviours were explored in-depth using a qualitative interview approach and have identified antihypertensive side effects to be a factor of non-adherence in pregnant women.
► About two-fifths of women who participated in this study were from Black, Asian and minority ethnic groups, providing a diverse range of voices.
► Respondents to the survey were self-selecting and may represent a relatively interested group of healthcare professionals.

## BACKGROUND

Hypertension in pregnancy is one of the leading causes of maternal mortality worldwide[1] and although mortality is declining in the UK,[2] women can still experience substantial morbidity from complications such as eclampsia and stroke.[3] Additionally, perinatal mortality remains high, with the UK population attributable risk of stillbirth from chronic hypertension at 14%[4] and around half of all neonates born to mothers who have had severe hypertension in pregnancy being admitted to the neonatal unit.[5] The morbidity and mortality attributable to hypertension, in many cases, may be modifiable through optimal use of antihypertensive agents during pregnancy.

The National Institute for Health and Care Excellence (NICE) hypertension in pregnancy guidelines (2010)[6] and linked quality statements (2013)[7] contain a quality statement regarding the provision of information on the use of safe antihypertensive medication in pregnancy and has related guidance that recommends discontinuation of teratogenic medications such as ACE inhibitors or angiotensin II receptor blockers with

prescribing of safe alternatives. Any prescribing of alternative antihypertensive medication should be dependent on prepregnancy treatment, side effect profiles and teratogenicity. A second quality statement advocates that women taking antihypertensive medication should have a blood pressure target (usually of less than 150/100 mm Hg) set in pregnancy. All NICE guidelines are underpinned by the recommendation of enabling patients to actively participate in their care which includes adopting a shared decision-making approach to treatment decisions.[8]

Despite publication of the guideline almost a decade ago, the implementation and evaluation of associated determinants of uptake have not been nationally evaluated. As a result, targeted strategies to reduce maternal and perinatal morbidity (and mortality) resulting from severe hypertension remain unidentified. Using the Consolidated Framework for Implementation Research (CFIR),[9] the aim of the study was to evaluate the implementation of NICE hypertension in pregnancy guidelines, to identify strategies to reduce incidence of severe hypertension and associated maternal and perinatal morbidity and mortality in pregnant women with chronic hypertension. In many countries, there is a movement toward establishing consensus-driven standardised clinical guidelines with the aim of improving patient safety and clinical outcomes. While new research continually emerges, guidelines are periodically updated and therefore remain an appropriate standard for evaluating routine clinical practice.[10]

## RESEARCH DESIGN AND METHODS
### Study setting and overall methodology
The Chronic Hypertension in pregnAncy iMPlementatION (CHAMPION study) is a multiple methods evaluation of the implementation of the NICE hypertension in pregnancy guidelines (2010 and updated in 2013) in women with chronic hypertension diagnosed before 20 weeks.[6 7] The study aimed to evaluate the variability in implementation of hypertension management practices set out in the NICE hypertension in pregnancy guidelines (2010).[6] As all guidelines should be underpinned by the 'Patient experience in adult National Health Service (NHS) services guideline',[8] which includes actively involving patient in decisions about their care through information provision and shared decision-making, the provision of information and women's involvement in decision making was also evaluated. The involvement of women in decision making was considered integral to the implementation study because successful hypertension management strategies involve the adherence to, alongside the prescribing of, antihypertensive medication.

Implementation was assessed through multiple methods: an online national survey of healthcare professionals, designed to describe general trends in guideline implementation; through review of the maternity case notes of women who had already given birth, a method that assessed the documentation of hypertension

management occurrence in each woman's maternity record. Aspects of care that would not normally be documented or are more difficult to capture, such as in-consultation discussions and occurrence of shared decision making were assessed through observations carried out by a midwife researcher (RW). The evaluation of the barriers and facilitators to implementation of NICE guidelines was assessed through qualitative interviews (with the same women and healthcare professionals who participated in the observation phase) using the CFIR. The study draws on CFIR as a theoretical framework to guide data collection, analysis and interpretation. The CFIR framework specifically evaluates five key domains that influence implementation; each domain has several subgroups to it, although only those relevant to this study have been identified. These include the intervention characteristics (the NICE guidelines), the outer context (the pregnant women), the inner context (NHS maternity services), individual context (the healthcare professionals) and the process of implementation (potential strategies).

Implementation of guidelines was assessed between November 2017 to December 2018 at three NHS Trusts with typical configurations of services for pregnant women with hypertension in the UK. Hospital Trust 1 was a tertiary city centre hospital with a newly formed specialist service that included consultant obstetricians, obstetric physicians and midwives who provided antenatal and intrapartum care to women with chronic hypertension within a specialist clinic; Hospital Trust 2 was a suburban district general hospital with a consultant-led antenatal clinic with antenatal midwives alongside providing care to women with a variety of pre-existing medical conditions; and Hospital Trust 3 had both a tertiary and a semi-rural hospital with a joint obstetric and physician led clinic and usual community-based midwifery care. No adjustment for clustering was required as no statistical comparison between sites was made. The NICE hypertension in pregnancy guidelines (2010)[6] had been adopted into local clinical guidelines at all three participating NHS Trusts for several years prior to the assessment of implementation.

### The National Survey
The implementation of evidence-based practices for the management of hypertension in pregnancy was assessed through self-reporting using an online survey (surveygizmo/s3). We embedded questions relating to the uptake of the NICE hypertension in pregnancy guidelines (2010)[6] using the 'template for intervention description and replication' (TIDieR) framework.[11] The 12-item TIDieR checklist (brief name, why, what (materials), what (procedure), who provided, how, where, when and how much, tailoring, modifications, how well (planned), how well (actual) is an extension of the Consolidated Standards of Reporting Trials 2010 statement (item 5) and the Standard Protocol Items: Recommendations for Interventional Trials 2013 statement (item 11). Although the emphasis of the TIDieR checklist is on reporting interventions for trials, the checklist was used

as a basis for this survey (but not as a reporting guideline) as it is also intended to apply across all evaluative study designs.[11] There is no single database of healthcare professionals' email addresses so national organisations including British Maternal and Fetal Medicine Society, Macdonald UK Obstetric Medicine Society and Royal College of Midwives were asked to email the survey (April to September 2018) to their members. No fee was charged as members' contact details were not shared with us and as a result the response rate could not be calculated. Ninety-seven healthcare professionals from 69 NHS Trusts responded, including 53 consultant obstetricians (55%), 16 doctors in training (16%), 22 specialist midwives (23%) and 6 community midwives (6%) (full copy of survey questions shown in online supplemental material 1).

## Case-notes review

The implementation of NICE guidelines (2010)[6] was also assessed through review of 100 maternity case notes of women with chronic hypertension identified from the electronic maternity records (32, 33, 35 women per Trust). At two of the Trusts, all women who had given birth in 2017 were included, whereas at the other Trust all women who had given birth over the final 3 months of 2017 were included as this third Trust had approximately four times the number of women with chronic hypertension per annum. In the UK, many women have abridged electronic maternity records and extensive handheld paper notes that are carried throughout pregnancy but are stored thereafter in the hospital. Both the electronic system and paper notes were obtained in the case notes review of care. Due to use of varying terms for hypertension on the electronic system, some women identified for case-note review were excluded as they did not have chronic hypertension when the full case notes were examined. Other reasons for exclusion included early miscarriage and transfer of care to another maternity unit. Data extraction based on the NICE hypertension in pregnancy guidelines (2010)[6] was completed by two midwife researchers (RW and HW), and minor discrepancies were resolved by discussion between the two researchers. It was not necessary to include a third reviewer as no major discrepancies were identified. Unclear or absent documentation including height, weight and body mass index (BMI) or antenatal blood pressure recordings was recorded as missing data. Severe hypertension was defined as systolic blood pressure greater than or equal to 160 mm Hg systolic or diastolic blood pressure greater than or equal to 110 mm Hg. For the assessment of BP targets, the quality statement related to documentation of a target (or not), not to the specific numerical thresholds chosen.

## Observations

Forty-two antenatal appointments involving 23 women with chronic hypertension and their respective doctors (nine) and midwives (five) were observed by a midwife researcher (RW) at the three NHS Trusts. Women with chronic hypertension were purposively sampled at their first obstetric antenatal appointment and, based on the availability of the midwife researcher, were approached consecutively along with their respective healthcare professionals until data saturation occurred. Staff and women gave written informed consent. Two women declined recruitment to the study. During observations, data about antenatal care provision were recorded using the Calgary-Cambridge communication guide[12] chosen for validity in relation to the research question, and its high inter-rater reliability. For example, offering choice is a subsection of shared decision making and is defined as 'encourages patient to make choices and decisions to the level that they wish'. Attainment of each section and subsections was established through the analysis of all 42 appointments using descriptive statistics.

## Semistructured interviews

Views about barriers and facilitators to implementation of evidence-based guidelines were collected from nine doctors and four midwives who were providing antenatal care for women with chronic hypertension. The interviews were carried out by a midwife researcher (RW) following informed consent and took place in privacy away from the clinical setting. The interviews were audio transcribed, coded and thematically analysed using inductive reasoning.[13] The codes generated formed small themes which were organised into the CFIR evaluation guide.[14] As formal implementation strategies had not been adopted beyond producing local guidance, interviewees were asked how they thought they could improve the implementation in the future.

Semistructured interviews with 18 women recruited for antenatal observations were carried out in the third trimester with informed consent. Women were asked about their antenatal care experiences using an interview schedule which reflected the concepts from the International Consortium for Health Outcome Measure (ICHOM) maternity standards sets[15] which include women's overall satisfaction with their care during pregnancy; satisfaction with information provision and their relationships with their care providers (see online supplemental material 2). ICHOM standards are internationally recognised measures that evaluate health outcomes that are important to patients (or pregnant women) and are used to improve local healthcare and compare outcomes internationally. The closed survey questions were turned into open ended questions to explore in-depth the quality of antenatal care provided. The interviews were carried out by a midwife researcher (RW) and took place away from the clinical setting, with assurance that discussions would not be shared with healthcare professionals and that participation or non-participation would not influence their care. The interviews were audio transcribed, coded and thematically analysed using an inductive approach. Women's experiences were analysed to improve understanding of their antenatal care needs,

which included how their hypertension was managed and the barriers and facilitators to the uptake of antihypertensives in pregnancy.

## Data analysis

The quantitative and qualitative data were analysed separately before being integrated. Descriptive analysis and summary statistics were used for the quantitative data. The semistructured interviews were thematically analysed by researchers (RW, JS and LC) using inductive techniques and typically lasted between 30 and 60 min.[16] The multiple methods data were integrated and analysed using the CFIR evaluation framework.[14] This included probing the inductively generated qualitative themes that related to implementation. The interpretation of the intervention constructs (characteristics, the inner and outer settings, the individual characteristics and the implementation processes) was carried out initially by the midwife researcher (RW) who collected the data, then with a second and third researcher (LC and JS) interpreting and discussing final interpretation of integrated data. Rigour was maintained through member reflection, attention to interview and transcription quality and systematic analysis. Rigour was improved using multiple data sources, a comprehensive integration framework (CFIR) and a multiple methods integration checklist.[17] Researchers were aware of, and sensitive to, the way in which their roles as midwives and doctor may have shaped the generation and analysis of the qualitative data.

## Patient and public involvement

A patient participant involvement (PPI) group consisting of women with experience of hypertension in pregnancy (n=7) and a maternity voices partnership group (n=15) provided feedback on the design of the study, research questions and outcome measures. The views of Black, Asian and minority ethnic women were purposively sought as they are disproportionately represented in the chronic hypertension in pregnancy population. PPI focus groups discussed what aspects of care were important to evaluate, this included the information women were given during pregnancy and whether women were involved in decisions about their care. They also provided constructively critical feedback on the patient information leaflets and consent forms.

## RESULTS

Antenatal care for women with chronic hypertension was provided by consultant obstetricians and midwives at all three hospitals. In two of the hospitals, women with chronic hypertension had designated midwives attached to the obstetric clinic. Approximately one-third of those recruited to the study had a BMI over 30 kg/m$^2$, approximately one-third were over the age of 35 and approximately two-fifths were of Black, Asian and minority ethnic backgrounds (shown in online supplemental material 3). Hospital Trust 1 had four times the population of women

with chronic hypertension compared with the other two units, comprising a large black minority ethnic population (many with associated comorbidities). Perinatal outcomes from the 55 pregnancies identified for case notes review showed that just under half of the women (46%) developed severe hypertension and that one in six babies were admitted to the neonatal unit (16%) (shown in online supplemental material 4). At all three hospitals medical history of women with chronic hypertension was inaccurate in the maternity records system and episodes of severe hypertension were recorded only in handwritten notes.

## Implementation of NICE hypertension in pregnancy 2010 guidelines and 2013 quality standards

### Setting a blood pressure target (quality statement 3)

Both the survey and the case notes review found the practice of setting an antenatal target blood pressure to be variable (table 1). Just over half of women with chronic hypertension had a target blood pressure documented in maternity notes (44% did not) yet substantial variation in practice between hospitals existed. At Hospital Trust 1, 77% of women had a target blood pressure documented in pregnancy compared with 23% and 38% at Hospital Trusts 2 and 3 respectively (online supplemental material 5). While it is possible that undocumented discussions occurred during consultations, which could not be extracted from case note review, such discussions would not be accessible on a longer-term basis to the woman or to other healthcare professionals involved in her care. The survey results support the case notes review findings as only a third of healthcare professional respondents reported always setting a target. The practice of undocumented 'unshared' target setting was identified through case notes review. Evidence of blood pressure targets being used by healthcare professionals but not shared with the woman and other professionals ('unshared') was frequently found. In about three-quarters of cases where the target blood pressure was unshared, and the blood pressure rose above systolic 150 mm Hg and or diastolic 100 mm Hg action was taken by professionals to lower it. Action was defined as making changes to blood pressure treatment, changing frequency of blood pressure monitoring or frequency of appointments (table 1).

Antihypertensive information provision, decision making and prescribing (quality statement 1 and associated guidance)

Variation in practice regarding first-line and second-line prescribing was identified through both the notes review and survey (table 1). In both, labetalol was the most commonly prescribed first line and nifedipine the most commonly used second-line antihypertensive agent; nevertheless, in about half of the case notes reviewed labetalol was not the first line antihypertensive prescribed. First line prescribing is not always exclusive as it may vary by ethnicity (eg, some doctors use labetalol as first line for many women, but nifedipine for Black women, in line with national guidelines for prescribing

**Table 1** Variation in implementation of evidence-based care evaluated through a national survey of obstetricians and midwives and women's case notes review at three representative NHS Trusts

| Care quality indicators | National survey n=97 (%) | Case notes review n=55 (%) |
|---|---|---|
| Blood pressure target setting (QS3) | | |
| Target blood pressure 'always' set | 36 (37.1) | |
| Target blood pressure 'almost always' set | 36 (37.1) | |
| Target blood pressure 'never' set | 1 (1.0) | |
| Target blood pressure not applicable (midwife) | 24 (23.3) | |
| Target blood pressure set at first opportunity (whichever first: booking or commencement of AHT) | – | 9 (18.0) |
| Target blood pressure not documented | | 26 (43.6) |
| Systolic target blood pressure | | |
| <160 mm Hg | 8 (8.2) | |
| <150 mm Hg | 89 (91.8) | 2 (7.4) |
| ≤140 mm Hg | | 27 (49.0) |
| Diastolic target blood pressure | | |
| <100 mm Hg | 94 (96.9) | 2 (7.4) |
| ≤90 mm Hg | | 27 (49.0) |
| Action taken to reduce blood pressure if above 150/100 mm Hg | | 13/17 (76.5) |
| Safe antihypertensive prescribing (linked to QS1) | | |
| ACEi and ARBs cessation | | |
| On ACEis or ARBs at antenatal booking appointment | | 4 (7.3) |
| Stopping ACEi or ARBs at first app if woman on either | | |
| Always | 57/86 (66.3) | – |
| Almost always | 27/86 (31.4) | – |
| ACEis or ARBs stopped at first obstetric appointment | | 4/4 (100.0) |
| First-line AHT prescribing (non-exclusive) | | |
| Labetalol | 85 (87.6) | 28 (50.9) |
| Nifedipine | 32 (33.0) | 9 (16.4) |
| Methyldopa | 29 (29.9) | 8 (14.5) |
| Other, for example, amlodipine | 2 (2.1) | 4 (7.3) |
| None | – | 6 (10.9) |
| Second line AHT prescribing (non-exclusive) | | |
| Nifedipine | 79 (81.4) | 9 (16.4) |
| Methyldopa | 60 (61.9) | 4 (7.3) |
| Labetalol | 38 (39.2) | 3 (5.4) |
| Amlodipine | 37 (38.1) | 2 (3.6) |
| Doxazosin | 23 (23.7) | 0 (0.0) |
| Other | 5 (5.2) | 0 (0.0) |
| None | – | 37 (67.3) |

ACEi, ACE inhibitors ; AHT, Anti-hypertensive; ARBs, Angiotensin II receptor blockers; NHS, National Health Service.

outside of pregnancy)[18] which may explain the variation in prescribing practice that existed (online supplemental material 5). Variation may also be explained by clinician preference or medication preference identified through shared decision making.

### Information provision about antihypertensive prescribing
Across all three Trusts, 52% (41/79) of the time the correct type and amount of information was provided

during the consultation (measured using the Calgary-Cambridge Guide). Visual techniques such as drawing or using charts to provide information occurred during consultation in 14% (3/21) of cases.

### Achieving a shared understanding: incorporating the woman's perspective
Of the survey respondents 96.9% strongly agreed or agreed that involving women with chronic hypertension

in management plans during pregnancy was important. However, when asked to give examples of how they involve women, only 4.3% identified discussing risks and benefits of treatment choice and 10% of respondents identified that women could be involved in plans about antihypertensive prescribing. The observations in the three hospital trusts found that 43% of the time (41/96) shared decision making occurred and 19% of women (3/16) were offered a choice regarding their hypertensive plans (including choice of antihypertensive).

## Barriers and facilitators to implementation (CFIR)
### Intervention characteristics (evidence and guideline)
All professionals interviewed, except one, saw value in having national guidance and understood that the local guidelines had been adapted from the 2010 national guideline.[6] Midwives relied more on local guidelines compared with obstetricians who referred more commonly to NICE guidelines. Some of the medical professionals had been involved in the development of an NICE guideline and were aware of the strengths and limitations of producing evidence-based guidelines in terms of the need for timely updating. Professionals described difficulties in creating guidelines where there is a paucity of robust data as is sometimes the case in maternity care. Weak, out of date or absent evidence influenced doctors' decisions not to implement guidelines. Some doctors described the weaknesses in the evidence underpinning the hypertension guidelines and described relying more on recent research compared with older national guidelines (table 2). The professionals identified that further research is necessary to support evidenced-based national guidelines (figure 1).

### Inner setting (organisation structure and culture)
The most frequently cited barriers to implementing high quality care for women with chronic hypertension were linked to the structure and organisation of antenatal care. Interviewees reported that a lack of consensus and guidance exists relating to models of care (such as whether specialist services would improve outcomes through better implementation) and pathways of care (such as frequency of blood pressure and medication reviews) (table 2). Evidence-based recommendations on models, and pathways of care, were identified as future facilitators to providing optimal antenatal care (figure 1). While most healthcare professionals initially described the uptake of the guidelines as a clinical priority during the interviews, clinicians identified difficulty with keeping up with recommendations and using them alongside clinical judgement as barriers to implementation (table 2).

Healthcare professionals considered the absence of written information a barrier to the uptake of antihypertensives in women with hypertension (table 2). A degree of paternalism exists in relation to involving women in decisions about their care. In principle, clinicians would like to involve women in decision making, yet they gave many examples of situations where they would exercise restraint in doing so (table 2). Education and tools to support shared decision making were identified as facilitators to optimising antenatal care for women with hypertension (figure 1).

### Characteristics of individuals (beliefs, knowledge and self-efficacy)
Interview analysis identified doctors' and midwives' knowledge and beliefs as the second most frequently cited barrier and facilitator to the implementation of hypertension management guidelines (table 2). There existed confusion about whether the guidelines sanction one antihypertensive medication over another for the management of chronic hypertension and if so, what evidence was used to support this. Likewise, confusion about blood pressure targets was described frequently as outcomes from a recent randomised controlled trial superseded the predated national guidelines (table 2). While midwives experienced less self-efficacy than the doctors, doctors still experienced difficulties in this area. They occasionally described the women's beliefs and views as a barrier to implementing the recommendations (table 2).

### Outer setting (women's views and experiences)
The quality of antenatal care experience was affected by women's internal conflict. There was also a high degree of variability in medication adherence (defined as, a blanket term factoring the extent to which patients' drug dosing histories conform, or not, to their corresponding prescribed drug dosing regimen)[19] and concordance (defined as, an agreement after negotiation between a woman and a healthcare professional that respects the beliefs and wishes of the woman in determining whether, when, and how medicines are to be taken).[20] Analysis identified that women require quality information about antihypertensives and their side effects, blood pressure ranges in pregnancy, as well as support to actively participate in decision making.

### Internal conflict
The majority (14 of 18) of women experienced internal conflict relating to the management of their hypertension during pregnancy, defined as a state of uncertainty about the course of action to take often in relation to making choices involving risk or uncertainty of outcomes (8) (figure 2A). The causes of internal conflict were identified as a lack of information provision, poorly managed side effects, women's personal beliefs and factors relating to the healthcare professional (table 3).

### Concordance
All women identified as concordant with healthcare professional management plans described being adherent to their antihypertensives. Facilitators to concordance included trust in the healthcare professional, mediated through information about safety of antihypertensives in pregnancy, knowledge about target blood pressure in pregnancy hypertension, acknowledgement of medication side-effects and a positive

**Table 2** Barriers to healthcare professional's implementation of hypertension in pregnancy guidelines, based on Consolidated Framework for Implementation Research (CFIR) implementation themes

| CFIR implementation themes | Frequency | Codes | Representative answer |
|---|---|---|---|
| **Intervention characteristics** | | | |
| Evidence strength, quality, source, and adaptability | 17 | AHT prescribing; target setting | ► 'I think the fact that it says use labetalol first line is not what we do, I don't believe the evidence for labetalol being better than methyldopa is there.'H <br> ► 'We can't get away from the fact that there aren't the source data there to make evidence-based guidelines.'B <br> ► 'So, I kept a close track of what was happening with the CHIPS study…I got a lot of information and knowledge from it.'A |
| **Inner setting** | | | |
| Structural characteristics | 43 | Information provision; pathways and models; training and education; time | ► 'I don't think we have a hand-out for, to give to hypertensive women about hypertension in pregnancy'.L <br> ► 'We don't have a dedicated hypertension clinic here. So, most of these women will get seen in general antenatal clinic'.I <br> ► 'You have people coming in three times weekly or something for their blood pressure, really? And other people who perhaps aren't being seen enough'.I |
| Relative priority | 26 | Guidelines; self-study; beliefs; experience | ► 'Well actually I don't even know what the NICE guidelines are for hypertension, I'm not a… as my colleagues will tell you, not a huge fan of NICE, in many ways.'L <br> ► 'I'm not just interested in guidelines; I'm interested in people's clinical experience…and that feel.'C |
| Culture of decision making | 19 | Patriarchy; shared decision making; type of decision: emergency, urgent and non-urgent | ► 'Doctors… see it as patients not doing what they're told'.A <br> ► 'I think that there's a balance to be had between involving women in the decisions, vs, them coming for expert recommendations'F <br> ► 'If I have a clinical situation where I want to start antihypertensives because she's got a dangerously high blood pressure, then that discussion is inevitably truncated.'B |
| **Individual characteristic** | | | |
| Beliefs about the intervention | 35 | AHT medication; AHT safety and side effects; target setting | ► 'National guidelines do not sanction any particular antihypertensive, or that the, the drug licenses do not sanction any particular antihypertensive'B <br> ► 'I think that might be something we're not quite as good at as we should be about defining a target for women….I suspect it's something we don't really document and clarify'H |
| Self-efficacy | 17 | Women's concordance/desire for involvement/first language | ► 'I think sometimes women don't necessarily want to make the decision'D <br> ► 'There's a lot of 'mumsnet'….and I would say they take a, that advice just as seriously as they do the advice that we give them here.'C |
| **Process of implementation** | | | |
| Engaging people and process of implementation | 16 | Using guidelines; updates, toolkits, and information; shared decision making | ► 'Awareness for people, if you're a busy jobbing healthcare practitioner, keeping up to date with each new area'H <br> ► 'Practical toolkits to help with that consultation'B <br> ► 'Evidenced based information having it more readily available for patient'D |
| Opinion leaders; Champions; | 5 | Utilisation of opinion leaders/champions in implementation | ► 'I find as a midwife sometimes you're a bit powerless, you know what the guidelines are, but depending on the doctor you're working with, tends to be the influencing factor on the decisions that are made… so it seems to be clinician-based guidelines sometimes, rather than the trust or national guidelines'D |

Letters[A-M] represent the healthcare professionals interviewed.
NICE, National Institute for Health and Care Excellence.

interaction with the healthcare professional (including communication and approach to decision making) (figure 2B).

## Adherence

Internal conflict was an important determinant of non-adherence (figure 2A) as only the women who expressed

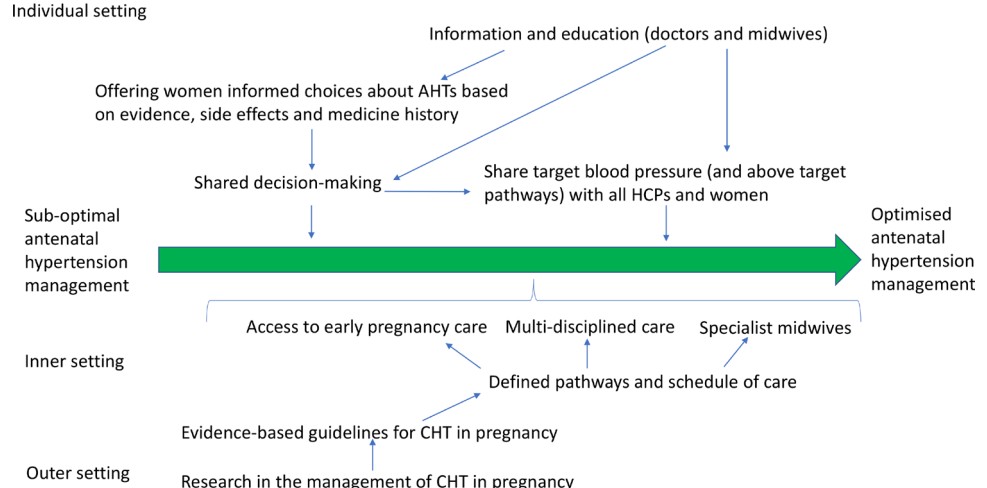

**Figure 1** Interpretation of integrated analysis: a strategy for improved implementation of evidence-based hypertension in pregnancy management. AHT, Anti-hypertensive; CHT, Chronic hypertension; HCP, healthcare professional.

internal conflict reported non-adherence to antihypertensive medication. Around half (8 of 18) the women interviewed described non-adherence to prescribed antihypertensives at some point during pregnancy with three women non-adherent at the time of interview (third trimester). However, 9 of 14 women describing internal conflict were adherent at the time of interview which was mediated by the 'responsibility of motherhood' rather than concordance with the hypertension management plan (figure 2B).

### Process of implementation (implementation strategies)

All three Trusts had a consultant obstetrician who led the care of women with chronic hypertension and could be considered the opinion leader. Two of three Trusts had a named midwife or team of midwives who specialised in the care of these women and were potential champions. However, influencers and champions were not always utilised to support guideline implementation. Further, as

implementation of the guidelines had not been audited in any of the Trusts, although some outcome data was routinely collected and analysed, opportunities to address unwanted variability were being missed. These findings are supported by the national survey which found only a quarter of the Trusts collected and analysed the outcomes of women with chronic hypertension in pregnancy.

### DISCUSSION

Women in this study (14/18) reported conflict relating to the uptake of prescribed antihypertensives in pregnancy and in many cases (8/14) internal conflict resulted in non-adherence. The most commonly cited reasons for conflict were lack of information provision, the side effects experienced from the medication, beliefs about safety of medication and uncertainty about normal blood pressure ranges in pregnancy. Adherence to antihypertensives in conflicted pregnant women was mediated

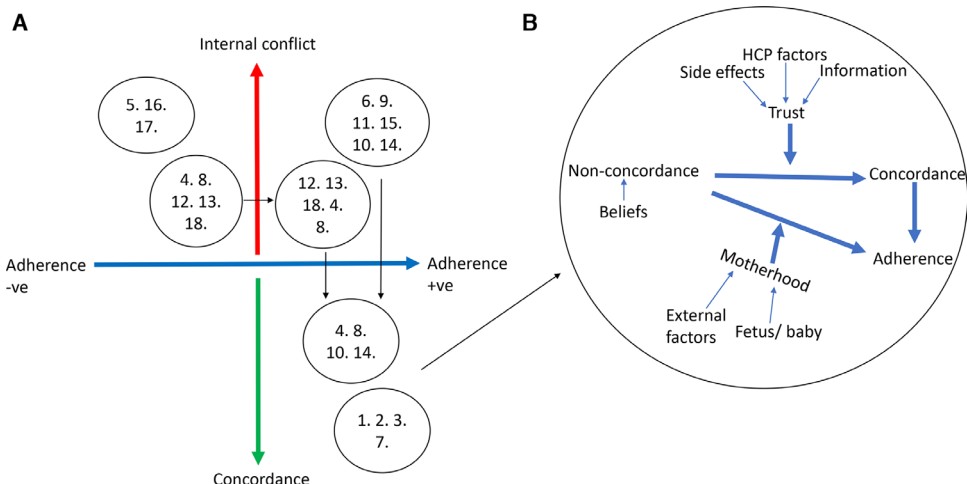

**Figure 2** (A) Women's adherence and concordance with prescribed antihypertensives. Numbers 1–18 represent interviewed women and their experiences of antihypertensive prescribing during pregnancy. Women who experienced a change in their adherence or in the reporting of internal conflict are plotted more than once in different bubbles. (B) Facilitators of women's adherence and of concordance. HCP, healthcare professional.

**Table 3** Barriers to women's uptake of hypertension in pregnancy guidelines

| CFIR outer context themes— Women's internal conflict | Frequency | Codes | Representative answer |
|---|---|---|---|
| Information | 30 | Medication (choices, dose, effectiveness, safety, interactions); severity of HTN; effect of HTN on pregnancy | '(I wanted to know) how safe it is, about the dosage, about the, taking the med-, this medication, about the side-effects and so and so and so, if they think any other option for me, or if this medication is not working, what will be the other option for me'J<br>'He was, you still need to carry on with your ramipril. I know I can't take it. It says in the leaflet not to take once you've hit 6 weeks, you need to stop. So, he was like oh, and then he phoned here, and he said oh well just take what you took before'H |
| Side effects | 21 | Maternal side effects; fetal side effects; Interactions; allergies; choices | 'They gave me first three, twice a day, then I was so giddy where I couldn't, if I take, I had to sleep all day for 2 days…Then I complained, but they still say to still take tablet.'I<br>'I'm on 18 pills a day, I do worry a bit about how they kind of potentially interact with each other and affect the baby'F |
| Beliefs | 17 | Hypertension status; understanding HTN; effectiveness AHT; safety AHT | 'I felt like I had to justify why I wasn't taking my tablet, which to me didn't seem right, 'cause if it, if my blood pressure was normal, and I took a tablet, surely my blood pressure then would be low?'Q<br>'cause everything I take my baby takes. So, it's like, what happens if my child comes out and then they're addicted to something, or they're high-strung because of something, or they're really moody and they're crying all the time because of the medicine I've had to take for the past 4 months'L |
| HCP factors | 17 | Continuity; listening to women; explaining regimes, mutual trust; communication | 'My issue has been where I've seen somebody who doesn't know the history, and typically they are a more junior doctor, and typically they are ticking a box and following a flow chart….the doctor said, you know, we're going to come to an agreement together but there was absolutely no discussion, she had no interest in what I had to say.'K |
| External factors | 7 | Family and friends; internet; access to services | 'My dad had been on beta blockers, which is what labetalol is, when he was younger, and he found, he was very ill on them, so he gave me a really negative impression of them'P |

LettersA-R represent the pregnant women interviewed.
AHT, Anti-hypertensive; CFIR, Consolidated Framework for Implementation Research; HCP, healthcare professional; HTN, Hypertension.

through a responsibility to motherhood rather than through a trusting partnership with healthcare professionals (supported by information provision, management of side effects and relational factors) as found in concordant adherent women. Despite this, our findings demonstrated that optimal information provision about antihypertensives and shared decision making occurred infrequently during antenatal consultations. Our findings also illustrated that the implementation of blood pressure target setting was suboptimal as a result of 'unshared' or undocumented target setting and in some cases an absence of target setting.

A major strength of the study is the recruitment of Black, Asian and minority ethnic women to both the research (40%) and in the PPI planning stage as these women are disproportionately represented in the chronic hypertension in pregnancy population. A further strength is the use of multiple methodological approaches and an implementation framework in order to improve reliability, validity and generalisability. However, results from the national survey may overstate compliance with national guidance. The survey was sent out to healthcare professionals from professional organisations; respondents

were, therefore, self-selecting and may represent a relatively interested group of healthcare professionals. The non-response rate is also unknown. The structured observations were carried out using a validated tool with high inter-rater reliability.[12] However, the observations were carried out by one midwife researcher which may affect the validity of the findings. Finally, the purposive sampling of healthcare professionals providing routine antenatal care for women with chronic hypertension resulted in a focus on lead carers (consultant obstetricians, obstetric medicine specialists and named midwives) being interviewed, rather than doctors in training and midwives in acute areas such as the maternity assessment unit.

The emergence of implementation science in recent years has identified that a gap between research findings and clinical practice exists, and that clinical guideline production does not ensure evidence-based practices are routinely adopted.[21] A recent study in British Colombia evaluated the implementation of recently published pregnancy hypertension guidelines and its associated effect on maternal and perinatal outcomes.[22] Following guideline dissemination the study reported a fall of about a third in combined adverse maternal health outcomes

(3.1%–1.9%) but did not report a significant reduction in adverse perinatal outcomes.[22] However, the wanted and unwanted variability in guidance uptake was not reported and the underlying mechanisms that influenced outcomes is not described. Our study uses an implementation framework by which variability in the implementation of existing guidelines could be described and mechanisms that support and hinder their uptake can be analysed, uniquely identifying strategies to improve the uptake of guidance and reduce maternal and fetal morbidity. Critically, although the NICE hypertension in pregnancy guidelines[6] have been recently updated, the core hypertension management recommendations remain unchanged, as do the quality statements. Therefore, the findings of this study remain important and relevant to those wanting to improve implementation.

The study also adds to the small body of antihypertensive adherence in pregnancy research that has found antihypertensive side effects are a determinant of non-adherence. One recent randomised controlled trial identified 11% of those included in randomisation discontinued the antihypertensive due to side effects.[23] Through the qualitative interview approach that enabled in depth exploration of women's medication behaviours, our study found about 40% of all women did not adhere to their prescribed antihypertensives at some point during pregnancy. This number compared more similarly to an internet-based study of 210 pregnant women undertaken in Europe, America and Australia which identified a 32.9% non-adherence rate in women taking cardiovascular medications in pregnancy.[24] These findings are supported by similar smaller questionnaire-based studies of pregnant women's medication adherence.[25 26] Our study may have identified higher rates of non-adherence due to the nature of qualitative interviewing that explore in-depth women's experiences and therefore unpick medication behaviours in a way that quantitative studies cannot.

Women's adherence to antihypertensives in pregnancy was found to be suboptimal, and strategies to improve adherence are likely to reduce incidences of severe hypertension and prevent associated morbidity (and mortality).[27] These include improved information provision about antihypertensives and blood pressure targets as well as embedding shared decision making into practice. Improvements in target blood pressure setting practices overall are also likely to reduce incidences of severe hypertension and prevent associated morbidity (and mortality).[3 5]

This study adds to the body of research that already exists outside of pregnancy which demonstrates that implementation of guidelines is not optimally achieved through the process of diffusion.[21] Although there was some evidence that some aspects of implementation were improved by having a specialist service for hypertension, this is likely to be most easily justified in areas where there is a high prevalence of chronic hypertension. Therefore, strategies to improve implementation in wider settings are required.

Professionals require guideline updates, implementation toolkits (to improve target blood pressure setting practices, standardised information about antihypertensives and in consultation aids) as well as support to have better conversations with their patients about medication choices and to improve the involvement of the women in the decision making. Professionals also need to buy into the evidence that underpins the guidance. Maternal and perinatal outcomes, which includes episodes of severe hypertension, should be collected annually, and used to support informed discussions about optimising antenatal care for this group of women.

Further research into the effectiveness and long-term safety of common antihypertensives in pregnancy and breast feeding to support evidenced-based guidelines is required.[28] Future research may also wish to evaluate strategies to reduce women's conflict regarding their antihypertensive use in pregnancy and establish the effect of interventions on maternal concordance and health outcomes. However, without further evidence relating to the safety and effectiveness of common antihypertensives it is unclear if further reductions in maternal and fetal morbidity can be achieved through prescribing practices. Future research should also focus on active implementation of blood pressure target setting and pathways for those with outside of target blood pressure readings. This is likely to reduce morbidity as target blood pressure setting in pregnancy has been shown to reduce incidences of severe hypertension.[3 5] Policy-makers may also wish to consider further studies that identify effective models and pathways of care for reducing adverse perinatal outcomes within the context of pregnancy hypertension.

## CONCLUSION

Maternal and neonatal morbidity resulting from severe hypertension in pregnancy is prevalent.[1 4 5] This evaluation of the implementation of the NICE hypertension in pregnancy guidelines (2010)[6] addresses strategies to reduce the number of episodes of severe hypertension and has identified suboptimal target setting practices, poor information provision for pregnant women and variability in prescribing practices. Women's non-adherence to antihypertensives is higher than previously reported and this is likely to be contributing to adverse perinatal outcomes. Analysis of the domains that influence implementation of the guidelines have identified that education and decision-making strategies are needed to address both clinician and women's behaviour. Further research into the effectiveness and long-term safety of common antihypertensives is also required.

**Contributors** RW, LC and JS conceived of the study, the manuscript and analyses, with contributions from LW, JG, HB and HW. RW was responsible for data management and data analysis. All authors reviewed, critically revised and approved the manuscript.

**Funding** This work was supported by the National Institute for Health Research (Research Professorship RP-2014-05-019) and by the National Institute for Health

Research (NIHR) Collaboration for Leadership in Applied Health Research and Care South London (NIHR CLAHRC South London) at King's College Hospital NHS Foundation Trust. Jane Sandall is an NIHR Senior Investigator and is supported by the National Institute for Health Research (NIHR) Applied Research Collaboration South London (NIHR ARC South London) at King's College Hospital NHS Foundation Trust.

**Disclaimer** The views expressed are those of the author(s) and not necessarily those of the NIHR or the Department of Health and Social Care.

**Competing interests** None declared.

**Patient consent for publication** Not required.

**Ethics approval** Ethical approval for the CHAMPION study was provided by the National Research Ethics Service (17/LO/2041).

**Provenance and peer review** Not commissioned; externally peer reviewed.

**Data availability statement** All data relevant to the study are included in the article or uploaded as online supplemental information. All data relevant to the study are included in the article.

**ORCID iD**
Rebecca Whybrow http://orcid.org/0000-0001-7484-1492

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
