## [Reviewer comments · BMJ Open]

ARTICLE DETAILS

TITLE (PROVISIONAL)	Implementation of national antenatal hypertension guidelines: a multi-centre multiple methods study
AUTHORS	Whybrow, rebecca; Webster, Louise; Girling, Joanna; Brown, Heather; Wilson, Hannah; Sandall, Jane; Chappell, Dr Lucy;

VERSION 1 – REVIEW

REVIEWER	Andrea Kattah Mayo Clinic, Rochester, MN, United States
REVIEW RETURNED	21-Jan-2020

GENERAL COMMENTS	This is a mixed methods study looking at the implementation of national hypertension in pregnancy guidelines. The NICE guidelines for hypertension in pregnancy in 2010 and a linked quality statement published in 2013 talked about the prescribing of anti-hypertensives and the importance of setting a blood pressure target in women taking anti-hypertensive medication. All of the guidelines endorse the importance of shared decision-making approach. The authors found a large degree of medication non-adherence in women with hypertension and BP control was sub-optimal. There was little documentation of choice and shared decision-making. The study is important in pointing out ways we can do better in caring for this high risk group and I liked that they tackled the question from multiple fronts. I have a few comments for the authors to consider: 1. The stated aim of this study was to evaluate the variance in the provision of and the barriers involved in the delivery of these national guidelines. However, through the interviews they conducted and their ensuing discussion, they have found that guidelines are sometimes out of date (example the CHIPS study showed that a lower BP target is safe) and based on poor evidence. Given the limitations of our knowledge and the difficulty for major guidelines to adapt to a rapidly changing literature base, are national guidelines really the metric by which to judge the provision of care? On page 10, line 356, the authors mention 'compliance with national guidance'. I am not aware that adherence to hypertension in pregnancy guidelines, per se, have been associated with better outcomes. As the authors point out, the study in British Columbia did not describe any mechanisms and did not show any difference in perinatal outcomes. I think something like the CHIPS trial, a large, well-performed, highly publicized RCT, will do more to change practice than guidelines that lag (example – SPRINT after AHA/ACC guidelines came out). I do not argue with the idea that treating BP in pregnancy to an appropriate target BP and shared-decision making are good goals, but rather whether adherence to an out-of-date guidelines is the metric to look at.
--

	2. Assuming that there are significant time constraints in clinic visits in the UK, is it possible in the chart review that discussions about targets, medications, and shared decision-making were had but not documented appropriately? Is this what you mean by undocumented unshared target setting? 3. If the survey was online, why was it sent to so few providers? Was a fee charged for their contact information? Also, why was the non-response rate unknown? 4. I would make it clearer earlier in the discussion that the study focused on women with chronic hypertension diagnosed before 20 weeks, not gestational hypertension or preeclampsia. 5. Please define severe hypertension (I assume it was 160/110) in the methods section. By not electronically recorded, does that mean that women had a high BP documented, but no note of it was made by the provider? Also, I assume that these higher readings were not taken during active labor where pain and other factors may play a role. 6. On page 7, line 227, when you discuss that 24% of providers did not take action on BP > 150/100, does that mean increasing medications or could they have the patient monitor home readings or return for more frequent measurements as a potential intervention? 7. Despite the limitations of guidelines noted in point 1, I completely agree with the authors conclusions that making better tools for counseling, particularly for providers that are not seeing the high volumes of patients and may not be up to date on all the recent data, better research and constant evaluation of maternal and perinatal outcomes, is essential moving forward. 8. I think the supplementary file Table on the demographics of the women would be reasonable to contain in the main manuscript.
--	--

REVIEWER	Gema Sanabria Universidad de Castilla-La Mancha, Spain
REVIEW RETURNED	09-Feb-2020

GENERAL COMMENTS	Actually, pregnancy hypertension is a prevalent problem with important negative outcomes in mothers and their offspring. Therefore, this is a relevant paper. Although it could has been a tedious explanation, the paper clearly guides us in how different methods have been used. But, I recommend that review the following issues of the paper:  • Lines 13-15 and 49-52: the aim of the study is different in both locations of the paper (abstract and introduction section). • Lines 53 and 77: The bibliography numbers are in different positions. So, authors must review this mistake throughout all document. • Data analysis section: Authors need explain with more detail and provide more information the analysis methods. • Discussion section: There are some methodological limitations that you could have solved in advance, such us sample size among other issues.
---

REVIEWER	Hazel Inskip University of Southampton, UK
REVIEW RETURNED	07-Apr-2020

GENERAL COMMENTS	This paper reports on a mixed methods approach to assessing the implementation of guidelines for managing hypertension in pregnancy. This is a useful study and it has highlighted that there are many challenges that need addressing. The study has some
--

limitations, but mainly these are clearly expounded by the authors, but it does highlight the challenges there are in implementing guidelines and it makes a good contribution to our understanding. I don't have major criticisms but I did find there were areas that I found hard to follow or where information appeared to be missing. I'm sorry if my comments appear picky but I hope that if the authors could consider these, it would improve clarity for the reader.

Abstract.

1. Line 17. The methods do not specify who was in the national survey. It needs to state that it was a survey of healthcare professionals
2. Line 27. In the results, I was unclear what 'conflict in taking antihypertensives' meant. In the paper it appears that it is 'internal conflict' that is being considered, not, for example, conflict between patient and healthcare professional

Background

1. Lines 73-77 should be in the methods I think.
2. Line 77 and elsewhere. As a statistician. I was puzzled by the use of the term 'variance'. Although correct linguistically, as 'variance' is so widely used in the literature as a statistical measure, I wondered whether 'variability' might be a better term.

Research design and methods

I had a number of points about the description of the methods:

1. Line 130. Could the authors explain the convenience sample? Did they select from the responses in some way? Or was this just the total number of responses from the mass mail out? If the latter then it's not really a convenience sample, it's just the response number. This is addressed a bit in the discussion line 357 and it appears to me that it is simply the response number.
2. Line 144. I think this information should be in the results not the methods. However, it looks very odd that the proportion of notes seen in Trust 1 are almost complete (29/32) but the proportions in Trusts 2 and 3 are well below 50%. This is not explained anywhere. Maybe it needs a note in the discussion?
3. Line 150. How were the women chosen? Were they consecutive women recruited and interviewed to saturation of themes? Were they opportunistic? Were they determined by shifts of the staff? An explanation of the recruitment is needed here, I think.
4. Line 160. Were the nine doctors and four midwives for the semi-structured interviews mainly the same as in the 'observations'? Or were they independent? More generally, are all the participants in each of the component sub-studies (national survey, case reviews, observations and the semi-structured interviews) all different from each other? I assume that the 18 women in the semi-structured interviews could have received their care from the doctors/midwives who also took part in semi-structured interviews, or were they completely independent people?
5. Line 184-5. What is meant by 'independent' results here? Why would they not be independent? Is it more that they were analysed separately by different people to avoid bias in the interpretation (particularly interpretation of the qualitative aspects if the national survey results were already known before the qualitative data were analysed)?

Results

1. Line 229. I struggled with some aspects of Table 1.
 - a. In the first section of the table, I would have expected for the

	national survey that the total of 97 should fit into the first three rows, but the total is only 72. Either woman were always set a target, sometimes or never/not-documented. Where are the missing 25? b. For the case notes review these top rows don't seem appropriate. We need to know how many had targets set and how many didn't. It's not apparent until one looks at the actual blood pressure results in the subsequent sections in the table that the remainder had targets set. Maybe separate the rows for the case note review here from the national survey? c. Surely the first line prescribing is exclusive, it's the second line prescribing that's not. 2. Lines 236-7. The implication of this statement is that labetalol should be the first line prescription. This is not mentioned in the introduction, but it is indicated as being the case by a comment in Table 2. For those not familiar with the guidelines this needs to be made clear. If it's not supposed to be the first line choice then perhaps this should be re-phrased 3. Line 263. There needs to be a 'not' before 'to' at the end of the line? 4. Lines 264-6. The reference to table 2 at the end of this sentence implies that the results only relate Trust 1, but I don't think that's the case. 5. Lines 266-7. It's not clear to me that what's described in this sentence is actually demonstrated in Figure 2, but Figure 2 is confusing to me anyway (see below). 6. Figure 2 labelling is confusing as it has labels 1a and 1b at the top of each section. I really don't understand Figure 2a at all. What are the numbers in the boxes? What does it mean? 7. Line 305. As noted above when commenting on the abstract, I think that the reference to conflict should always be to internal conflict. Conflict on its own is confusing and could imply conflict between health professionals and the pregnant women. 8. Line 408. I was surprised that there is no mention of supporting health professionals to have better conversations with their patients about medication choices and to improve the involvement of the women in the decision-making. This seemed to be a strong message coming out of the paper to me. Supplementary file 1. 1. Presumably fourth line of first section of the table should say Northern Ireland not just Ireland. The NHS doesn't operate south of the border. 2. Q 8, needs the word 'scan' added to the question after 'fetal' 3. Q12 SDM and IOL should be spelled out. Supplementary file 2. Are those who were interviewed a subset of the observed? I've queried this point in comment 4 in the research design and methods section. Supplementary file 4. This table needs to specify that it relates to the case note review (or at least that's what I think it relates to).
--	--

REVIEWER	Angela Lupattelli University of Oslo, Norway
REVIEW RETURNED	30-Apr-2020

GENERAL COMMENTS	This is an interesting study and the authors have collected unique datasets using a multi-centre mixed methods approach. The paper is
---

	well written and structured. The authors have done a good job in regards to integration of the qualitative and quantitative data. However, the integrated data has more to offer to address the research question and inform understanding of the barriers and facilitators to implementation of the guidelines. Please find below some comments and suggestions. Methods: Could the authors expand the description about the sampling of the three hospital trusts? I feel more elaboration on this point is needed. Also, data from each hospital may be clustered and therefore non-independent. I wonder whether specific data analyses accounting for clustered data by hospital, were considered. I understand that the sample per hospital is small. However, as shown in the supplementary data, there were differences in prescribing across hospital trusts. Maybe the table presented in supplementary file 4 should be presented in the main article. Regarding the case-note review part of study: About half of the selected maternity records met the inclusion criteria. This seems quite high and should be commented. Could the authors also elaborate on why the aimed sample was set to 100 records? It seems the authors aim to extract quantitative data from this component of the study, but the number of records is so low that correct interpretation of these findings is difficult. Line 158-182: it would be nice to present some info regarding the research team and reflexivity (part of the COnsolidated criteria for REporting Qualitative research (COREQ) checklist) Please consider expanding on whether the women for semi-structured interviews were purposively sampled, and about the applied sampling procedure. It would be helpful to also state the time duration of the interviews. Line 186: The authors have mentioned that they analysed the interview data based on inductive techniques but was the analysis informed by any principle or authors' perspective as well? In addition, multiple readers/coders should be involved in the qualitative data analysis but this information is missing in the description. Line 193: analytic integration methods: What kind of integration methods were used in addition to the CFIR? Please elaborate.
--	---

REVIEWER	Dr Sabrina Grant University of Worcester, UK
REVIEW RETURNED	20-May-2020

GENERAL COMMENTS	This is a clearly well written and structured evaluation of the implementation of NICE guidance in pregnancy guidelines. Methods are robust and appropriate to the topic. Given the complex methodology involved in this evaluation it does feel a little limited in places with detail. My only comments relate to the methodology.
--

	The National Survey - is there more detail about the recruitment of these clinicians - other than from 69 trusts? that is, how the survey was distributed? - via email link? total number of clinicians emailed? Recruitment method is a bit limited. e.g. '(surveygizmo/s3) Case-notes review Again (32,33,35 per trust) - not sure what the means, clarify please. Is this the local trust code? Should discrepancies be resolved by a third reviewer? can more information be reported about missing data - and how this was handled? Semi-structured interviews CFIR evaluation guide is referenced. Do you have a reference of the thematic analysis methodology? Can you supply the topic guide as supplementary file to understand how the themes within the table are drawn. Table 3 at present isn't quite clear. For example what do you mean by Items? Do tables 2 and 3 relate to the implementation framework used? Have you got the tidier checklist as a supplementary file with further detail of these frameworks being applied. Strengths and Limitations - what is so novel about the qualitative interview approach used? I don't really understand the fourth limitation (population size in each of the methods) and not relevant for qualitative research. Refine. PPI - any further detail of numbers of individuals in the PPI group - were they just involved at the beginning then for informing design of the study? What activities were undertaken during these sessions. Again a bit limited in detail with such an extensive mixed methods study.
--	--

VERSION 1 – AUTHOR RESPONSE

We have indicated edits in red text. Line numbers refer to the track changes version of the manuscript uploaded.

Reviewer Name: Andrea Kattah	
1a. The stated aim of this study was to evaluate the variance in the provision of and the barriers involved in the delivery of these national guidelines. However, through the interviews they conducted and their ensuing discussion, they have found that guidelines are sometimes out of date (example the CHIPS study showed that a lower BP target is safe) and based on poor evidence. Given the limitations of our knowledge and the difficulty for major guidelines to adapt to a rapidly changing literature base, are national guidelines really the metric by which to judge the provision of care?	a. We have edited the manuscript as follows (line 87) Using the Consolidated Framework for Implementation Research (CFIR), the aim of the study was to evaluate the implementation of NICE hypertension in pregnancy guidelines, to identify strategies to reduce incidence of severe hypertension and associated maternal and perinatal morbidity and mortality in pregnant women with chronic hypertension. In many countries, there is a movement toward establishing consensus-driven standardised clinical guidelines with the aim of improving patient safety and clinical

	outcomes. Whilst new research continually emerges, guidelines are periodically updated and therefore remain an appropriate standard for evaluating routine clinical practice.
1b. On page 10, line 356, the authors mention 'compliance with national guidance'. I am not aware that adherence to hypertension in pregnancy guidelines, per se, have been associated with better outcomes. As the authors point out, the study in British Columbia did not describe any mechanisms and did not show any difference in perinatal outcomes. I think something like the CHIPS trial, a large, well-performed, highly publicized RCT, will do more to change practice than guidelines that lag (example – SPRINT after AHA/ACC guidelines came out).	b. We have included a sentence in the Introduction (as detailed in our response to 1a above) to make this clearer.
1c. I do not argue with the idea that treating BP in pregnancy to an appropriate target BP and shared-decision making are good goals, but rather whether adherence to an out-of-date guideline is the metric to look at.	c. The guidelines cover many aspects of care, of which treating BP to a target is just one. We were assessing whether any target was set, not the exact numbers (i.e. whether to 2010 guidelines or to CHIPS practice). We have edited the manuscript to make this clearer (line 180): Data extraction based on the NICE hypertension in pregnancy guidelines (2010)⁶ was completed by two midwife researchers (RW, HW), and discrepancies were resolved by discussion between the two researchers. Unclear or absent documentation was recorded as missing data. Severe hypertension was defined as systolic blood pressure greater than or equal to 160 mmHg systolic or diastolic blood pressure greater than or equal to 110 mmHg. For the assessment of BP targets, the quality statement related to documentation of a target (or not), not to the specific numerical thresholds chosen.
2a. Assuming that there are significant time constraints in clinic visits in the UK, is it possible in the chart review that discussions about targets, medications, and shared decision-making were had but not documented appropriately?	We have edited the manuscript as follows (line 275) Both the survey and the case-notes review found the practice of setting an antenatal target blood pressure to be variable (table 1). Just over half of women with chronic hypertension had a target blood pressure documented in maternity notes (44% did not) yet substantial variation in practice

	between hospitals existed. At Hospital Trust 1, 77% of women had a target blood pressure documented in pregnancy compared to 23% and 38% at Hospital Trusts 2 and 3 respectively (supplementary material 4). Whilst it is possible that undocumented discussions occurred during consultations, which could not be extracted from case-note review, such discussions would not be accessible on a longer term basis to the woman or to other healthcare professionals involved in her care. The survey results support the case-notes review findings as only
2b. Is this what you mean by undocumented unshared target setting?	We have edited the original section in the manuscript as follows (line 282): The survey results support these findings as only a third of healthcare professional respondents reported always setting a target. The practice of undocumented 'unshared' target setting was identified through case-notes review. Evidence of blood pressure targets being used by healthcare professionals but not shared with the woman and other professionals ('unshared') was found. In about three quarters of women whose blood pressure rose above systolic 150mmHg and or diastolic 100mmHg action was taken by professionals to lower it (table 1).
3. If the survey was online, why was it sent to so few providers? Was a fee charged for their contact information? Also, why was the non-response rate unknown?	We have edited the manuscript as follows (line 145) Although the emphasis of the TIDieR checklist is on reporting interventions for trials, the checklist was used as a basis for this survey (but not as a reporting guideline) as it is also intended to apply across all evaluative study designs. There is no single database of healthcare professionals' email addresses so national organisations including British Maternal and Fetal Medicine Society (BMFMS), Macdonald UK Obstetric Medicine Society (MOMS) and Royal College of Midwives (RCM) were asked to email the survey (April to September 2018) to their members. No fee was charged as members' contact details were not shared with us and as a result the response rate could not be calculated. Ninety-seven healthcare professionals from sixty-nine NHS Trusts was obtained, including 53 consultant obstetricians (55%), 16 doctors in training (16%), 22 specialist midwives (23%) and six community midwives (6%) (full copy of survey questions shown in

	supplementary material 1)
4. I would make it clearer earlier in the discussion that the study focused on women with chronic hypertension diagnosed before 20 weeks, not gestational hypertension or preeclampsia.	We have edited the manuscript as follows (line 94) The CHAMPION study (Chronic Hypertension in pregnancy iMPlementatIOn study) is a multi-methods evaluation of the implementation of the NICE hypertension in pregnancy guidelines (2010 and updated in 2013) in women with chronic hypertension diagnosed before 20 weeks.
5a. Please define severe hypertension (I assume it was 160/110) in the methods section.	a. We have edited the manuscript as follows (line 184) Severe hypertension was defined as systolic blood pressure greater than or equal to 160 mmHg systolic or diastolic blood pressure greater than or equal to 110 mmHg. For the assessment of BP targets, the quality statement related to documentation of a target (or not), not to the specific numerical thresholds chosen.
5b. By not electronically recorded, does that mean that women had a high BP documented, but no note of it was made by the provider?	b. We have edited the manuscript to as follows (line 267) At all three hospitals, medical history of women with chronic hypertension was inaccurate in the maternity records system and episodes of severe hypertension were recorded only in handwritten notes.
5c. Also, I assume that these higher readings were not taken during active labor where pain and other factors may play a role.	We have edited the manuscript as follows, to indicate that this is a study of antenatal hypertension, not covering intrapartum care (lines 1 and 13): Title: Implementation of national antenatal hypertension guidelines: a multi-centre mixed methods study Objective To evaluate the implementation of NICE antenatal hypertension guidelines, to identify strategies to reduce incidences of severe hypertension and associated maternal and perinatal morbidity and mortality in pregnant women with chronic hypertension.
6. On page 7, line 227, when you discuss that 24% of providers did not take action on BP > 150/100, does that mean increasing medications	We have edited the manuscript as follows to make the actions clearer (line 288):

or could they have the patient monitor home readings or return for more frequent measurements as a potential intervention?	Action was defined as making changes to blood pressure treatment, changing frequency of blood pressure monitoring or frequency of appointments (table 1).
7. Despite the limitations of guidelines noted in point 1, I completely agree with the authors conclusions that making better tools for counseling, particularly for providers that are not seeing the high volumes of patients and may not be up to date on all the recent data, better research and constant evaluation of maternal and perinatal outcomes, is essential moving forward.	We agree and have aimed to cover this in the discussion.
8. I think the supplementary file Table on the demographics of the women would be reasonable to contain in the main manuscript.	We agree that the table is interesting but due to restrictions on the number of tables we were unable to include it in the main body of the manuscript. We are happy to be led by editors regarding its inclusion.

Reviewer Name: Gema Sanabria	
 • Lines 13-15 and 49-52: the aim of the study is different in both locations of the paper (abstract and introduction section). 	We have edited the manuscript as follows to ensure that the aims match (line 78): Using the Consolidated Framework for Implementation Research (CFIR), the aim of the study was to evaluate the implementation of NICE hypertension in pregnancy guidelines, to identify strategies to reduce incidence of severe hypertension and associated maternal and perinatal morbidity and mortality in pregnant women with chronic hypertension.
 • Lines 53 and 77: The bibliography numbers are in different positions. So, authors must review this mistake throughout all document. 	We think that the reviewer is referring to the additional space between the punctuation mark and the reference numbers, as shown below (lines 59 and 82): and stroke.3 collected.9 We have checked the manuscript throughout and corrected this.
 • Data analysis section: Authors need explain with more detail and provide more information the analysis methods. 	We have provided more detail and made clearer the methods section of the manuscript. We have made track changes to the study setting and overall methodology, the survey, the case-notes review, the observations, the data analysis and

	the patient participation involvement.
• Discussion section: There are some methodological limitations that you could have solved in advance, such as sample size among other issues.	We have edited the manuscript as follows (line 190) Forty-two antenatal appointments involving 23 women with chronic hypertension and their respective doctors (nine) and midwives (five) were observed by a midwife researcher (RW) at the three NHS Trusts. Women with chronic hypertension were purposively sampled at their first obstetric antenatal appointment and based on the availability of the midwife researcher, were approached consecutively along with their respective healthcare professionals until data saturation occurred. Staff and women gave written informed consent. Two women declined recruitment to the study. Additionally, we have edited the manuscript as follows (line 148) There is no single database of professionals' email addresses so national organisations including British Maternal and Fetal Medicine Society (BMFMS), Macdonald UK Obstetric Medicine Society (MOMS) and Royal College of Midwives (RCM) were asked to email the survey (April to September 2018) to their members. No fee was charged as members' contact details were not shared with us and as a result the response rate could not be calculated. Ninety-seven healthcare professionals from sixty-nine NHS Trusts was obtained, including 53 consultant obstetricians (55%), 16 doctors in training (16%), 22 specialist midwives (23%) and six community midwives.

Reviewer Name: Hazel Inskip	
Abstract.	
1. Line 17. The methods do not specify who was in the national survey. It needs to state that it was a survey of healthcare professionals	We have edited the manuscript as follows (line 17): We used a national survey of healthcare professionals (n=97)
2. Line 27. In the results, I was unclear what 'conflict in taking antihypertensives' meant. In the paper it appears that it is 'internal conflict' that is	We have edited the manuscript as follows (line 29):

being considered, not, for example, conflict between patient and healthcare professional	Women (14/18) reported internal conflict in taking antihypertensives
Background	
1. Lines 73-77 should be in the methods I think.	This has been moved to line 116 in methods. The manuscript has been edited as follows (line 112) The evaluation of the barriers and facilitators to implementation of NICE guidelines was assessed through qualitative interviews (with women and healthcare professionals) using the consolidated Framework for Implementation Research (CFIR). The study draws on CFIR as without a theoretical framework to guide data collection, analysis, and interpretation.
2. Line 77 and elsewhere. As a statistician. I was puzzled by the use of the term variance'. Although correct linguistically, as 'variance' is so widely used in the literature as a statistical measure, I wondered whether 'variability' might be a better term.	We have edited the manuscript throughout, replacing variance with variability in order to avoid use of a statistical term (variance).
Research design and methods	
I had a number of points about the description of the methods: 1. Line 130. Could the authors explain the convenience sample? Did they select from the responses in some way? Or was this just the total number of responses from the mass mail out? If the latter then it's not really a convenience sample, it's just the response number. This is addressed a bit in the discussion line 357 and it appears to me that it is simply the response number.	We have edited the manuscript as follows (line 148) There is no single database of professionals' email addresses so national organisations including British Maternal and Fetal Medicine Society (BMFMS), Macdonald UK Obstetric Medicine Society (MOMS) and Royal College of Midwives (RCM) were asked to email the survey (April to September 2018) to their members. No fee was charged as members' contact details were not shared with us and as a result the response rate could not be calculated. Ninety-seven healthcare professionals from sixty-nine NHS Trusts was obtained, including 53 consultant obstetricians (55%), 16 doctors in training (16%), 22 specialist midwives (23%) and six community midwives.
2. Line 144. I think this information should be in the results not the methods. However, it looks very odd that the proportion of notes seen in Trust 1 are almost complete (29/32) but the proportions in Trusts 2 and 3 are well below 50%. This is not	We have edited the manuscript as follows Removed line 144 and amended line 264 as follows

explained anywhere. Maybe it needs a note in the discussion?	Perinatal outcomes from the fifty-five pregnancies identified for case-notes review showed that just under half of the women (46%) We have edited the manuscript as follows (line 159) The implementation of NICE guidelines (2010)⁶ was also assessed through review of 100 maternity case-notes of women with chronic hypertension identified from the electronic maternity records (32, 33, 35 women per Trust). At two of the Trusts all women who had given birth in 2017 were included, whereas at the other Trust all women who had given birth over the final three months of 2017 were included as this third Trust had approximately four times the number of women with chronic hypertension per annum. In the UK, many women have abridged electronic maternity records and extensive handheld paper notes that are carried throughout pregnancy but are stored thereafter in the hospital. Both the electronic system and paper notes were obtained in the case-notes review of care. Due to use of varying terms for hypertension on the electronic system, some women identified for case-note review were excluded as they did not have chronic hypertension when the full case-notes were examined. Other reasons for exclusion included early miscarriage and transfer of care to another maternity unit.
3. Line 150. How were the women chosen? Were they consecutive women recruited and interviewed to saturation of themes? Were they opportunistic? Were they determined by shifts of the staff? An explanation of the recruitment is needed here, I think.	We have edited the manuscript as follows (line 190) Forty-two antenatal appointments involving 23 women with chronic hypertension and their respective doctors (nine) and midwives (five) were observed by a midwife researcher (RW) at the three NHS Trusts. Women with chronic hypertension were purposively sampled at their first obstetric antenatal appointment and based on the availability of the midwife researcher, were approached consecutively along with their respective healthcare professionals until data saturation occurred. Staff and women gave written informed consent. Two women declined

	recruitment to the study.
4. Line 160. Were the nine doctors and four midwives for the semi-structured interviews mainly the same as in the 'observations'? Or were they independent? More generally, are all the participants in each of the component sub-studies (national survey, case reviews, observations and the semi-structured interviews) all different from each other? I assume that the 18 women in the semi-structured interviews could have received their care from the doctors/midwives who also took part in semi-structured interviews, or were they completely independent?	We have edited the manuscript as follows, addressing the reviewer's queries (line 106): Implementation was assessed through multiple methods: an online national survey of healthcare professionals, designed to describe general trends in guideline implementation; through review of the maternity case-notes of women who had already given birth, a method that assessed the documentation of hypertension management occurrence. Aspects of care that would not normally be documented or are more difficult to capture, such as in-consultation discussions and occurrence of shared decision-making were assessed through antenatal observations carried out by a midwife researcher (RW). The evaluation of the barriers and facilitators to implementation of NICE guidelines was assessed through qualitative interviews (with the same women and healthcare professionals recruited to the observation phase) using the Consolidated Framework for Implementation Research (CFIR). We have edited the supplementary file 3 as follows: Maternal demographics of women observed, interviewed and included for case-note review. Women interviewed are a subset of those observed. Case-notes identified for review are a different cohort of women.
5. Line 184-5. What is meant by 'independent' results here? Why would they not be independent? Is it more that they were analysed separately by different people to avoid bias in the interpretation (particularly interpretation of the qualitative aspects if the national survey results were already known before the qualitative data were analysed)?	We have edited the manuscript as follows (line 227): The quantitative and qualitative data were analysed separately before being integrated.
Results	
1. Line 229. I struggled with some aspects of Table 1.	We have addressed this point by point below.

1a. In the first section of the table, I would have expected for the national survey that the total of 97 should fit into the first three rows, but the total is only 72. Either woman were always set a target, sometimes or never/not-documented. Where are the missing 25?	a. We have edited the manuscript as follows (table 1 line 292) Target setting not applicable (midwife) 24 (23.3)
1b. For the case notes review these top rows don't seem appropriate. We need to know how many had targets set and how many didn't. It's not apparent until one looks at the actual blood pressure results in the subsequent sections in the table that the remainder had targets set. Maybe separate the rows for the case note review here from the national survey?	b. We have edited the manuscript as follows (table 1 line 292) Target blood pressure 'never' set 1 (1.0) Target blood pressure not documented 26 (43.6)
1c. Surely the first line prescribing is exclusive, it's the second line prescribing that's not	c. We have edited the manuscript as follows (line 296) Variation in practice regarding first- and second-line prescribing was identified through both the notes review and survey (table 1). In both, labetalol was the most commonly prescribed first line and nifedipine the most commonly used second line antihypertensive agent; nevertheless, in about half of the case-notes reviewed labetalol was not the first line antihypertensive prescribed. First line prescribing is not always exclusive as it may vary by ethnicity (e.g. some doctors use labetalol as first line for many women, but nifedipine for Black women, in line with national guidelines for prescribing outside of pregnancy) which may explain the variation in prescribing practice that existed when comparing different hospital Trusts (supplementary material 4). Variation may also be explained by clinician preference or medication preference identified through shared decision-making.
2. Lines 236-7. The implication of this statement is that labetalol should be the first line prescription. This is not mentioned in the introduction, but it is indicated as being the case by a comment in Table 2. For those not familiar with the guidelines this needs to be made clear. If it's not supposed to be the first line choice then perhaps this should be re-phrased	We have edited the manuscript as shown in the box above.
3. Line 263. There needs to be a 'not' before 'to'	We have edited the manuscript as follows (line

at the end of the line?	329): Weak, out of date or absent evidence influenced doctors' decisions notto implement guidelines.
4. Lines 264-6. The reference to table 2 at the end of this sentence implies that the results only relate Trust 1, but I don't think that's the case.	We have edited the manuscript as follows (line 329) Weak, out of date or absent evidence influenced doctors' decisions to implement guidelines. Some doctors interviewed described the weaknesses in the evidence underpinning the hypertension guidelines and described relying more on recent research compared to older national guidelines (table 2).
5. Lines 266-7. It's not clear to me that what's described in this sentence is actually demonstrated in Figure 2, but Figure 2 is confusing to me anyway (see below).	We have clarified that the sentence that refers to figure 1, which includes the statement 'Research in the management of cronic hypertension in pregnancy'
6. Figure 2 labelling is confusing as it has labels 1a and 1b at the top of each section. I really don't understand Figure 2a at all. What are the numbers in the boxes? What does it mean?	We have edited the manuscript to ensure figure 2a. and 2b. are labelled as such. A legend has been added to figure 2a. (line 608) Numbers 1-18 represent interviewed women and their experiences of anti-hypertensive prescribing during pregnancy. Women who experienced a change in their adherence or in the reporting of internal conflict are plotted more than once in different bubbles.
7. Line 305. As noted above when commenting on the abstract, I think that the reference to conflict should always be to internal conflict. Conflict on its own is confusing and could imply conflict between health professionals and the pregnant women.	We have edited the manuscriptreplacing 'conflict' with 'Internal conflict' throughout
8. Line 408. I was surprised that there is no mention of supporting health professionals to have better conversations with their patients about medication choices and to improve the involvement of the women in the decision-making. This seemed to be a strong message coming out of the paper to me.	We have edited the manuscript as follows (line 481) Professionals require guideline updates, implementation toolkits (to improve target blood pressure setting practices, standardised information about antihypertensives and in consultation aids)as well as support to have better conversations with their patients about medication choices and to improve the involvement of the women in the decision-making.Professionals also need to buy into the evidence that underpins the guidance.Maternal and perinatal outcomes, which

	includes episodes of severe hypertension, should be collected annually and used to support informed discussions about optimising antenatal care for this group of women.
Supplementary file 1.	
1. Presumably fourth line of first section of the table should say Northern Ireland not just Ireland. The NHS doesn't operate south of the border.	We have edited the manuscript as follows Northern Ireland
2. Q 8, needs the word 'scan' added to the question after 'fetal'	We have edited the manuscript as follows Fetal growth scans
3. Q12 SDM and IOL should be spelled out.	We have edited the manuscript as follows Shared decision-making Induction of labour
Supplementary file 2.	
Are those who were interviewed a subset of the observed? I've queried this point in comment 4 in the research design and methods section.	We have edited the manuscript as follows (line 112) The evaluation of the barriers and facilitators to implementation of NICE guidelines was assessed through qualitative interviews (with the same women and healthcare professionals recruited to the observation phase) using the Consolidated Framework for Implementation Research (CFIR). We have edited the manuscript supplementary file 2 as follows Women interviewed are a subset of those observed. Case-notes identified for review are a different cohort of women.
Supplementary file 4.	
This table needs to specify that it relates to the case note review (or at least that's what I think it relates to).	We have edited the manuscript as follows Target blood pressure setting and prescribing practices per Trust – as derived from case-note review

Reviewer Name: Angela Lupattelli	
Methods: 1a. Could the authors expand the description about the sampling of the three hospital trusts? I feel more elaboration on this point is needed. Also, data from each hospital may be clustered and therefore non-independent. I wonder whether specific data analyses accounting for clustered data by hospital, were considered. I understand that the sample per hospital is small. However, as shown in the supplementary data, there were differences in prescribing across hospital trusts. 1b. Maybe the table presented in supplementary file 4 should be presented in the main article	1a. We have edited the manuscript as follows (line 132) ;and Hospital Trust 3 had both a tertiary and a semi-rural hospital with a joint obstetric and physician led clinic and usual community-based midwifery care. No adjustment for clustering was required as no statistical comparison between sites was made. 1b. We agree that the table is interesting but due to restrictions on the number of tables we were unable to include it in the main body of the manuscript. We are happy to be led by editors regarding its inclusion.
2a. Regarding the case-note review part of study: About half of the selected maternity records met the inclusion criteria. This seems quite high and should be commented. agree 2b. Could the authors also elaborate on why the aimed sample was set to 100 records? It seems the authors aim to extract quantitative data from this component of the study, but the number of records is so low that correct interpretation of these findings is difficult.	2a. We have edited the manuscript as follows (line 159) The guidelines (2010)⁶ was also assessed through review of 100 maternity case-notes of women with chronic hypertension identified from the electronic maternity records (32, 33, 35 women per Trust). At two of the Trusts all women who had given birth in 2017 were included, whereas at the other Trust all women who had given birth over the final three months of 2017 were included as this third Trust had approximately four times the number of women with chronic hypertension per annum. In the UK, many women have abridged electronic maternity records and extensive handheld paper notes that are carried throughout pregnancy but are stored thereafter in the hospital. Both the electronic system and paper notes were obtained in the case-notes review of care. Due to use of varying terms for hypertension on the electronic system, some women identified for case-note review were excluded as they did not have chronic hypertension when the full case-notes were examined. Other reasons for exclusion included early miscarriage and transfer of care to another maternity unit.

	We have edited the manuscript as follows Perinatal outcomes from the fifty-five pregnancies identified for case-notes review showed that just under half of the women (46%) developed severe hypertension and that one in six babies were admitted to the neonatal unit (16%) (shown in supplementary material 3). 2b. We have edited the manuscript as follows (line 159) The implementation of NICE guidelines (2010)⁶ was also assessed through review of 100 maternity case-notes of women with chronic hypertension identified from the electronic maternity records (32, 33, 35 women per Trust). At two of the Trusts all women who had given birth in 2017 were included, whereas at the other Trusts all women who had given birth over the final three months of 2017 were included as this third Trust had approximately four times the number of women with chronic hypertension per annum.
Line 158-182: it would be nice to present some info regarding the research team and reflexivity (part of the Consolidated criteria for Reporting Qualitative research (COREQ) checklist)	We have edited the manuscript as follows (line 238) Rigour was improved using multiple data sources, and analytic integration methods. a comprehensive integration framework (CFIR) and a mixed methods integration checklist. Researchers were aware of, and sensitive to, the way in which their roles as midwives and doctor may have shaped the generation and analysis of the qualitative data.
Please consider expanding on whether the women for semi-structured interviews were purposively sampled, and about the applied sampling procedure. It would be helpful to also state the time duration of the interviews.	We have amended the manuscript as follows (line 190) Forty-two antenatal appointments involving 23 women with chronic hypertension and their respective doctors (nine) and midwives (five) were observed by a midwife researcher (RW) at the three NHS Trusts. Women with chronic hypertension were sampled at their first hypertension clinic appointment. Women were recruited consecutively along with their respective healthcare professionals until data saturation occurred.

	We have edited the manuscript as follows (line 229) The semi-structured interviews were thematically analysed using inductive techniques and typically lasted between 30 and 60 minutes.
a.Line 186: The authors have mentioned that they analysed the interview data based on inductive techniques but was the analysis informed by any principle or authors' perspective as well? b.In addition, multiple readers/coders should be involved in the qualitative data analysis but this information is missing in the description.	We have edited the manuscript as follows (line 227): The quantitative and qualitative data were initially analysed separately to generate independent results. Descriptive analysis and summary statistics were used for the quantitative data. The semi structured interviews were thematically analysed by researchers (RW, JS and LC) using inductive techniques and typically lasted between 30 and 60 minutes. The mixed-methods data were integrated and analysed using the CFIR evaluation framework. This included probing the inductively generated qualitative themes that related to implementation further. The interpretation of the intervention constructs (characteristics the inner and outer settings, the individual characteristics and the implementation processes) was carried out initially by the midwife researcher (RW) who collected the data, then with a second and third researcher (LC, JS) interpreting and discussing final interpretation of integrated data.
Line 193: analytic integration methods: What kind of integration methods were used in addition to the CFIR? Please elaborate.	We have edited the manuscript as follows (line 238) Rigour was improved using multiple data sources, and analytic integration methods. a comprehensive integration framework (CFIR) and a mixed methods integration checklist. Researchers were aware of, and sensitive to, the way in which their roles as midwives and doctor may have shaped the generation and analysis of the qualitative data.

Reviewer Name: Dr Sabrina Grant	
1. The National Survey - is there more detail	We have edited the manuscript as follows (line

about the recruitment of these clinicians - other than from 69 trusts? that is, how the survey was distributed? - via email link? total number of clinicians emailed? Recruitment method is a bit limited. e.g. '(surveygizmo/s3)	145): Although the emphasis of the TIDieR checklist is on reporting interventions for trials, the checklist was used as a basis for this survey (but not as a reporting guideline) as it is also intended to apply across all evaluative study designs. There is no single database of healthcare professionals' email addresses so national organisations including British Maternal and Fetal Medicine Society (BMFMS), Macdonald UK Obstetric Medicine Society (MOMS) and Royal College of Midwives (RCM) were asked to email the survey (April to September 2018) to their members. No fee was charged as members' contact details were not shared with us and as a result the response rate could not be calculated. Ninety-seven healthcare professionals from sixty-nine NHS Trusts was obtained, including 53 consultant obstetricians (55%), 16 doctors in training (16%), 22 specialist midwives (23%) and six community midwives (6%) (full copy of survey questions shown in supplementary material 1)
Case-notes review 2a. Again (32,33,35 per trust) - not sure what the means, clarify please. Is this the local trust code? 2b. Should discrepancies be resolved by a third reviewer? 2c. can more information be reported about missing data - and how this was handled?	2a. We have edited the manuscript as follows (line 159) The implementation of NICE guidelines (2010) was also assessed through case-note review of 100 women with chronic hypertension identified in the maternity electronic databases (32, 33, 35 women per Trust). 2b. We have edited the manuscript as follows (line 180) Data extraction based on the NICE hypertension in pregnancy guidelines (2010) was completed by two midwife researchers (RW, HW), and minor discrepancies were resolved by discussion between the two researchers. It was not necessary to include a third reviewer as no major discrepancies were identified. 2c. We have edited the manuscript as follows (line 183) Unclear or absent documentation including height, weight and body mass index or antenatal

	blood pressure recordings were recorded as missing data
Semi-structured interviews 3a.CFIR evaluation guide is referenced. Do you have a reference of the thematic analysis methodology? 3b.Can you supply the topic guide as supplementary file to understand how the themes within the table are drawn. 3c.Table 3 at present isn't quite clear. For example what do you mean by Items? 3d. Do tables 2 and 3 relate to the implementation framework used? 3e.Have you got the tidier checklist as a supplementary file with further detail of these frameworks being applied.	We have edited the manuscript and included the following reference Braun, V. & Clarke, V. (2006). Using thematic analysis in psychology. Qualitative Research in Psychology, 3(2), 77-101 3b. We have included a supplementary file with the topic guides and renumbered the supplementary files accordingly. 3c. We have edited the manuscript and replaced table 2 and 3 items with codes 3d. We have edited the manuscript as follows Table 2 format Replaced barriers with CFIR implementation themes Table 3 Replaced Women'ssources ofconflict with CFIR outer context themes – Women's internal conflict 3e. We have clarified that we used the TIDIER framework to inform the survey questions, not as the reporting guideline for this manuscript. We have edited the manuscript as follows (line 145)Although the emphasis of the TIDieRchecklist is on reporting interventions for trials, the checklist was used as a basis for this survey (but not as a reporting guideline) as it is also intended to apply across all evaluative study designs.
4a.Strengths and Limitations - what is so novel about the qualitative interview approach used? 4b.I don't really understand the fourth limitation (population size in each of the methods) and not	4a. We have edited the manuscript as follows (line 461) Through the qualitative interview approach that enabled in depth exploration of women's medication behaviours, our study found about 40% of all women did not adhere to their

relevant for qualitative research. Refine.	prescribed antihypertensives at some point during pregnancy. 4b. We have edited the manuscript as follows We have removed the fourth limitation from both the abstract and from the discussion.
5. PPI - any further detail of numbers of individuals in the PPI group - were they just involved at the beginning then for informing design of the study? What activities were undertaken during these sessions. Again a bit limited in detail with such an extensive mixed methods study.	We have edited the manuscript as follows (line 243) A patient participant involvement (PPI) group consisting of women with experience of hypertension in pregnancy (n=7) and a south London maternity voices partnership group (n=15) provided feedback on the design of the study, research questions and outcome measures. The views of Black, Asian and minority ethnic women were purposively sought as they are disproportionately represented in the chronic hypertension in pregnancy population. PPI focus groups discussed what aspects of care were important to evaluate, this included the information women were given during pregnancy and whether women were involved in decision about their care. They also provided constructively critical feedback on the patient information leaflets and consent forms. We have also edited the manuscript as follows (line 428) A major strength of the study is the recruitment of Black, Asian and minority ethnic women to both the research (40%) and in the PPI planning stage as these women are disproportionately represented in the chronic hypertension in pregnancy population. A further strength is the use of multi-methodological approaches and an implementation framework in order to improve reliability, validity and generalisability. However, results from the national survey may overstate compliance with national guidance. We have also edited the manuscript as follows (line 50) About two-fifths of women who participated in this study were from Black, Asian and minority

	ethnic groups,providing a diverse range of voices. We have edited the manuscript as follows (line 259) approximately one-third were over the age of 35 and approximately two-fifths Black, Asian and minority ethnic backgrounds (shown in supplementary material 2).
--	--

VERSION 2 – REVIEW

REVIEWER	Andrea Kattah Mayo Clinic, United States
REVIEW RETURNED	15-Jul-2020

GENERAL COMMENTS	All of my comments are addressed. If possible, and acceptable to the editor, would recommend adding the demographic table to the supplemental material.
---

REVIEWER	Hazel Inskip University of Southampton
REVIEW RETURNED	20-Jul-2020

GENERAL COMMENTS	clearly to my comments, which I appreciate. The paper reads well. My comments now are really minor but, in all but one case, they are grammatical errors that have arisen due to an amendment elsewhere in a sentence. I hope these are helpful in finalising the manuscript. Line 13. Now that 'antenatal' is used in the objective in the abstract, the words 'in pregnancy' could do with deleting. Line 23. Replace 'principle' with 'principal'. Line 154. Replace 'was' with 'were' now that the changes have been made to that sentence. However I think it's odd to say the people were 'obtained'. Might it be better to say that they 'responded' instead? Line 192. Add a comma after 'and' and before 'based'. Line 248. Replace 'decision' with 'decisions'. Line 263. 'Pregnancies' needs to start with a capital letter. Line 287. The end of the sentence now does not make sense due to the amendments earlier in the sentence.
--

REVIEWER	Angela Lupattelli University of Oslo, Norway
REVIEW RETURNED	03-Jul-2020

GENERAL COMMENTS	Thank you for revising the manuscript, most of my comments were addressed. However I think there is a misunderstanding regarding my prior comment on clustered data by sites. Data clustering should be taken into account in the main analysis when total data across sites are analyzed. This is not something done specifically when comparisons are made between sites.
---

REVIEWER	Dr Sabrina Grant University of Worcester
REVIEW RETURNED	16-Jul-2020

GENERAL COMMENTS	Thank you for addressing the previous comments. My revisions point to the abstract - do you mean sub-optimal instead of sub-optional? 192 - should cite a page number if a direct quote I am presently uncertain as to whether this paper needs majorly revising in terms of structure. The main problem with this paper is that there are several evaluative methods used here which are quite separate in their own right. This is not quite the pure mixed methods study - but a study of multiple methods as described in the main body. Mixed methods and multiple methods are used interchangeably and should be consistent throughout. Presently the methodology section does not read fluently and needs to be more succinct in narrative to replicate. I am also not fond of writing in the first person i.e. 'We...!' which doesn't seem suitable for BMJ Open. Referring to my comments, the dealing of missing data is not adequate. This is not simply what missing data there was but by what methods were these handled during the data analysis. What statistical methods did you apply to account for the missing data. There are several additional edits about BAME populations and is now a strength of the study. I question why this was added retrospectively.
---

VERSION 2 – AUTHOR RESPONSE

Angela Lupattelli	
Data clustering should be taken into account in the main analysis when total data across sites are analyzed. This is not something done specifically when comparisons are made between sites.	We appreciate that in statistical analysis of large quantitative data sets clustering would be taken into account across all three sites and embedded into the main analysis. However, as the study uses a mixture of qualitative, observational and basic descriptive quantitative data that explores and describes the implementation of guidelines, adjustment for clustering has not been employed across the quantitative data set.
Andrea Kattah	
If possible, and acceptable to the editor, would recommend adding the demographic	We have clarified that the demographic table can be found in supplementary material three (line

table to the supplemental material.	233).
Dr Sabrina Grant	
do you mean sub-optimal instead of sub-optional?	We have edited the manuscript as follows: Sub-optimal information provision around treatment, choice of antihypertensives and target setting practices by healthcare professionals may be contributory.
192 - should cite a page number if a direct quote	We have edited the manuscript as follows: As all guidelines should be underpinned by the 'Patient experience in adult NHS services guideline' ⁸ which includes, actively involving patient in decisions about their care through information provision and shared decision-making, the provision of information and women's involvement in decision-making was also evaluated.
This is not quite the pure mixed methods study - but a study of multiple methods as described in the main body. Mixed methods and multiple methods are used interchangeably and should be consistent throughout.	We have edited the manuscript as follows: Multiple methods have replaced mixed methods to ensure consistency throughout.
Presently the methodology section does not read fluently and needs to be more succinct in narrative to replicate.	We appreciate that the method section is longer than some single method studies are. We have sought to extensively describe the methods undertaken in each of the multiple methods as well as describe how these methods have been integrated using a validated framework. The detailed nature of the methods section should enable better replication for those interested in this type of study.
I am also not fond of writing in the first person i.e. 'We....' which doesn't seem suitable for BMJ Open	We have reviewed the BMJ Open author guidelines and consider that the use of the first person "we" is suitable. We are aware that some writing styles use the passive voice, but other editors prefer the clarity of the first person. We are happy to take editorial advice.
the dealing of missing data is not adequate. This is not simply what missing data there was but by what methods were these handled during the data analysis. What statistical methods did you apply to account for the missing data.	Due to the descriptive nature of the paper statistical methods for handling missing data were not employed.
There are several additional edits about BAME populations and is now a strength of	The reviewers suggested we look again at the strengths and limitations of the study. Alongside

the study. I question why this was added retrospectively.	this we had several comments from reviewers to include the demographic data in the main body or highlight its relevance. This prompted us to include the inclusion of BAME women more prominently in the manuscript.
Reviewer: 3	
Line 13. Now that 'antenatal' is used in the objective in the abstract, the words 'in pregnancy' could do with deleting.	We have amended the manuscript as follows: To evaluate the implementation of NICE antenatal hypertension guidelines, to identify strategies to reduce incidences of severe hypertension and associated maternal and perinatal morbidity and mortality in pregnant women with chronic hypertension.
Line 23. Replace 'principle' with 'principal'.	We have edited the manuscript as follows principal
Line 154. Replace 'was' with 'were' now that the changes have been made to that sentence. However I think it's odd to say the people were 'obtained'. Might it be better to say that they 'responded' instead?	We have edited the manuscript as follows Ninety-seven healthcare professionals from sixty-nine NHS Trusts responded, including 53 consultant obstetricians
Line 192. Add a comma after 'and' and before 'based'.	We have edited the manuscript as follows appointment and, based ...
Line 248. Replace 'decision' with 'decisions'.	We have edited the manuscript as follows this included the information women were given during pregnancy and whether women were involved in decisions about their care
Line 263. 'Pregnancies' needs to start with a capital letter.	The clean version of the manuscript reads 'Perinatal outcomes from the fifty-five pregnancies identified for case-notes review...' and so we have left pregnancies non-capitalised.
Line 287. The end of the sentence now does not make sense due to the amendments earlier in the sentence.	We have edited the manuscript as follows: Evidence of blood pressure targets being used by healthcare professionals but not shared with the woman and other professionals ('unshared') was frequently found. In about three quarters of cases where the target blood pressure was unshared, and the blood pressure rose above systolic 150mmHg and or diastolic 100mmHg action was taken by professionals to lower it.